# Viral Innate Immune Evasion and the Pathogenesis of Emerging RNA Virus Infections

**DOI:** 10.3390/v11100961

**Published:** 2019-10-18

**Authors:** Tessa Nelemans, Marjolein Kikkert

**Affiliations:** Department of Medical Microbiology, Leiden University Medical Center, Leiden, 2333 ZA, The Netherlands; tessa-nelemans@hotmail.com

**Keywords:** positive-sense single-stranded RNA viruses, innate immune evasion, type I and III interferons, viral pathogenesis

## Abstract

Positive-sense single-stranded RNA (+ssRNA) viruses comprise many (re-)emerging human pathogens that pose a public health problem. Our innate immune system and, in particular, the interferon response form the important first line of defence against these viruses. Given their genetic flexibility, these viruses have therefore developed multiple strategies to evade the innate immune response in order to optimize their replication capacity. Already many molecular mechanisms of innate immune evasion by +ssRNA viruses have been identified. However, research addressing the effect of host innate immune evasion on the pathology caused by viral infections is less prevalent in the literature, though very relevant and interesting. Since interferons have been implicated in inflammatory diseases and immunopathology in addition to their protective role in infection, antagonizing the immune response may have an ambiguous effect on the clinical outcome of the viral disease. Therefore, this review discusses what is currently known about the role of interferons and host immune evasion in the pathogenesis of emerging coronaviruses, alphaviruses and flaviviruses.

## 1. Introduction

The positive-sense single-stranded RNA viruses (+ssRNA) comprise many pathogens that are a serious threat to human health. These viruses affect millions of people worldwide and for many of them, no treatment is available. Human +ssRNA viruses that currently have a large impact on public health include dengue virus (DENV) and the more recent (re-)emerging viruses such as chikungunya virus (CHIKV) and Zika virus (ZIKV) [1]. Although Middle East respiratory syndrome-coronavirus (MERS-CoV) does not infect large numbers of people like the afore-mentioned viruses, 35% of diagnosed patients die of MERS [2], and considering the virus’ genetic possibilities, any change that increases transmissibility between humans would mean a serious public health threat that is considered worth preparing for. Gaining insight into the pathogenesis of these viruses is thus relevant to thoroughly understand them and identify new treatment and prevention options.

Our innate immune system is the first line of defence against invading pathogens. It has developed multiple mechanisms to sense invading viruses, and subsequent signal transduction pathways are targeted at the initial containment of infection in the body. Conserved microbial structures, known as pathogen-associated molecular patterns (PAMPs), can be sensed by pattern recognition receptors (PRRs) of the innate immune system, resulting in the activation of downstream signalling pathways that then elicit an effective antimicrobial response [3]. Interferons (IFNs) are key molecules in this response. There are three families of interferons, of which type I and III IFNs are regarded as the main effectors during the antiviral innate response in many different cell types, while type II IFNs are more specifically expressed by specialized immune cells and are involved in the modulation of adaptive immunity. Both type I and III IFNs activate the so-called “interferon-stimulated genes” (ISGs), which promote an anti-viral state in the infected as well as in neighbouring cells, restrict viral replication, and induce apoptosis to protect the organism from virus spread [4,5]. Cytolytic +ssRNA viruses outrun this response through rapid transmission to new hosts [6], while other (noncytolytic) +ssRNA viruses, in order to create a window of opportunity for efficient replication despite the innate immune system, have developed multiple mechanisms to prevent IFN induction and subsequent signalling. Active viral innate immune evasion strategies have been shown to target PRRs, downstream signalling molecules and transcription factors of the IFN pathway, as well as its effector ISG products [7,8]. Additionally, a major more passive way in which +ssRNA viruses are thought to prevent detection by innate immune sensors is by hiding viral replication intermediates in rearranged intracellular membranes, the so-called replication organelles or factories [9,10].

One of the reasons why +ssRNA viruses are causing pathology in humans is probably the ability to actively evade the innate immune system and in particular the IFN response. The targeting of the interferon response by the +ssRNA viruses has been quite extensively reviewed [7,8,11,12,13,14,15]. The strategies of viral proteins to suppress IFN signalling are thus well established, although the diversity in the strategies used suggests that further research will most probably identify yet different evasion mechanisms. Importantly, most of these reviewed studies focused on in vitro experiments and/or they examined ectopically expressed proteins to determine the role of viral factors in innate immune evasion. The actual link between host innate immune evasion and the pathology observed during viral infection generally receives less attention and therefore remains to be comprehensively investigated [16]. In order to set the stage, we will here review and discuss the available data on this latter subject. We will discuss viral proteins targeting innate immunity from emerging viruses belonging to the Togaviridae, Coronaviridae and Flaviviridae families and summarize the available knowledge on the influence of these viral activities on the pathogenesis of the virus infections. The main focus of this review will be on describing the evasion of the type I and type III IFN response and its influence on infection outcome based on advances from studies conducted ex vivo or in vivo.

## 2. Innate Immune Response to Positive-Sense Single-Stranded RNA Viruses

During viral infection, type I and III IFNs are rapidly induced to initiate the antiviral state. The type I IFN family consists of 13 subtypes of IFNα (in humans), one subtype of IFNβ and several subtypes that are poorly studied (IFNκ, IFNɛ, IFNτ, IFNδ, IFNω and IFNζ) [5,17]. The more recently discovered type III IFN system consists of IFN-λ1, IFN-λ2, IFN-λ3 and IFN-λ4 [5]. The trigger for type I and III IFN production and the downstream signalling molecules are actually similar for the two types [18,19]. However, type III interferons preferentially act on epithelial cells providing antiviral protection locally and are thought to cause less immune-mediated damage than the more potent, systemic type I IFN response [20,21,22]. Therefore, it is now proposed that type III IFNs are an initial response to knock down infection at the epithelial barriers without causing immunopathology, while type I IFNs come into play when this first line of defence is not sufficient to control the infection [21]. The production of these two types of IFNs is initiated when the innate immune system recognizes invading viruses through its PRRs. Three types of PRRs sensing RNA viruses have been well described: the Toll-like receptors (TLRs), nucleotide-binding oligomerization domain (NOD)-like receptors (NLRs) and retinoic acid-inducible gene I (RIG-I)-like receptors (RLRs)[23]. Specifically, the activation of the IFN response through TLR3, TLR7, RIG-I and melanoma differentiation-associated protein 5 (MDA5) is well studied [3,23].

TLR3 is activated by double-strand RNA (dsRNA) and upon activation, it will signal through TRIF (TIR-domain-containing adapter-inducing interferon-β), which results in the induction of the following four transcription factors: activator protein 1 (AP-1), nuclear factor kappa-light-chain-enhancer of activated B cells (NF-κB) and interferon regulator factors 3 and 7 (IRF3 and IRF7). These four transcription factors are involved in the regulation of IFN expression, while only NF-κB and AP-1 are involved in the induction of pro-inflammatory cytokines [24]. The activation of the first two transcription factors is achieved by the recruitment of receptor-interacting protein 1 (RIP1) and TNF receptor-associated factors 2 and 6 (TRAF2/6). This pathway ultimately leads to the degradation of the inhibitor of kB (IκB) and thereby to the release of NF-κB. The activation of the other two transcription factors is achieved by signalling through TRAF3, which activates two kinases: TRAF family member-associated NF-κB activator binding kinase 1 (TBK1) and inducible IκB kinase ε (IKKε). Active TBK1 and IKKε can phosphorylate IRF3 and IRF7 to promote their translocation to the nucleus [3,23,24]. TLR7, which is located in the endosomes, recognizes ssRNA and signals through the myeloid differentiation primary response gene 88 (MyD88) pathway. MyD88 forms a complex with interleukin-1 receptor-associated kinases 1 and 4 (IRAK-4 and IRAK-1) and TRAF6. Next, this complex can activate the transcription factor IRF7 [3,23,24] (Figure 1A).

RIG-I and MDA5 can sense viral molecules in the cytoplasm. RIG-I exposes its caspase activation and recruitment domains (CARD) upon binding of short RNAs containing 5′-triphosphate ends and polyuridine-rich sequences [25]. CARD can bind to the adaptor protein mitochondrial antiviral signalling (MAVS), which interacts with TRAF3 and TRAF2/6, thereby converging on TLR3 pathways. The binding of RIG-I to MAVS is further promoted by polyubiquitination of the CARD domain and subsequent tetramerisation of RIG-I. As described above, TRAF3 triggers TBK1 and IKKε to phosphorylate IRF3 and IRF7, after which the transcription factors translocate to the nucleus. Signalling via TRAF2/6 results in the activation of NF-κB [24,26] (Figure 1A). MDA5 is thought to trigger the same signalling cascade as RIG-I, except that MDA5 is activated by long dsRNA [24,25,26]. IRFs are essential for the induction of IFNα/β and IFNλ, while NF-κB is thought to be required as a co-factor. The expression of interferons in the early stage of viral infection is mainly dependent on IRF3, which is constitutively expressed. IFNs then trigger the transcription of IRF7 which can serve to amplify the IFN response [5,24].

When IFNα/β and IFNλ are induced, they bind to their cognate receptors to induce the expression of ISGs. While IFNα/β bind to a heterodimeric receptor consisting of the subunits IFNAR1 and IFNAR2, IFNλ signals through a heterodimeric receptor composed of IFNLR1 and IL-10 receptor β (IL10RB) [5,27]. Remarkably, both receptors then activate the same signalling cascade, which led to the idea that the type III IFN system might be redundant. Nonetheless, there seems to be some difference in the expression levels and kinetics of the stimulated ISGs, even though the systems share the same signal transduction pathways [21,28]. Binding to type I IFN or type III IFN receptor leads to the assembly of the ISG factor 3 (ISGF3) complex. This complex is built up from the heterodimer signal transducer and activator of transcription 1 (STAT1), STAT2 and IRF9. STAT1 and STAT2 heterodimerization is initiated after they have been phosphorylated by Janus kinase 1 (JAK1) and tyrosine kinase 2 (TYK2). ISGF3 is capable of binding to the IFN-stimulated response elements in the ISG promoter, resulting in the expression of the ISGs that will induce the antiviral state [5,27] (Figure 1B). Together, the ISGs virtually act on every step in the viral life cycle (i.e., entry, replication, translation, assembly and budding) [29]. 

Similar to interferons, NF-κB can regulate the expression of hundreds of genes. NF-κB regulates genes that are involved in the recruitment of innate immune cells, in the activation of the pro-inflammatory state in these cells, as well as in the activation and differentiation of inflammatory T cells [30]. It is for example an important transcription factor for the production of pro-inflammatory cytokines including interleukin 1β (IL-1β), IL-6 and tumour necrosis factor α (TNF-α) and of chemokines such as IL-8, C–C motif chemokine ligand 2 (CCL2) and C–X–C motif chemokine 10 (CXCL10) [31]. NF-κB signalling is thus an important mediator in controlling virus infection but it has also been implicated in inflammatory diseases [32]. 

## 3. Coronaviruses

### 3.1. Genome Organization

The human coronaviruses were regarded as lowly pathogenic for a long time. However, the more recently discovered severe acute respiratory syndrome CoV (SARS-CoV) and MERS-CoV proved otherwise. The SARS-CoV outbreak in 2002–2003 caused a worldwide epidemic with a global case-fatality rate of around 11% [33]. The MERS-CoV epidemic that started in 2012 is still ongoing, with an estimated case-fatality rate of 35% globally [2]. Both of these emerging viruses can cause severe respiratory illness [34,35,36]. In addition, MERS-CoV can cause renal dysfunction. 

The coronaviruses have the largest known positive-stranded RNA genome of around 30 kb with a 5’-terminal cap and a poly(A) tail at the 3’ end [37] (Figure 2). The 5’-terminal two-thirds of the genome encode two large open reading frames (ORFs), 1a and 1b. The expression of the viral genome results in the production of two polypeptides (pp1a and pp1ab). The translation of pp1ab is accomplished by a −1 ribosomal frameshift at the end of ORF1a. These polypeptides are then processed by two or three internally virus-encoded proteases into 16 nonstructural proteins (nsps). The nsps are involved in the replication of the viral genome, in subgenomic mRNA synthesis, and also in the evasion of host innate immunity [37,38]. The remaining one-third of the genome encodes the viral structural proteins spike (S), envelope (E), membrane (M), and nucleocapsid (N) and accessory proteins. The accessory genes have also been implicated in antagonizing the IFN response. SARS-CoV encodes eight accessory proteins, while MERS-CoV encodes five accessory proteins [39].

### 3.2. The Role of Interferons in Coronavirus Pathogenesis

The IFN response is considered important for the control of coronavirus infection. IFN treatments improved the outcomes of SARS-CoV and MERS-CoV infection in mice and in non-human primates [40,41,42,43,44]. In human SARS-CoV-infected patients, IFNα therapy in combination with steroids was associated with higher oxygen saturation levels and faster improvement of lung abnormalities [45]. In patients with a MERS-CoV infection, treatment with IFNα in combination with ribavirin resulted in an improved survival rate at 14 days after diagnosis. However, this effect was not present anymore after 28 days, and other studies concluded that there is no beneficial effect of the used IFN treatments [46,47,48]. Currently, another trial to study the efficacy of IFNβ1b in combination with lopinavir/ritonavir in MERS patients is still ongoing [49]. Therefore, there is no actual proof of the efficacy of IFN treatment yet. Notwithstanding, the expression levels of IFN during the disease course do suggest that IFN is important. For example, studies with macaques showed that aged macaques infected with SARS-CoV had a more severe pathology compared to young adults, while the induction of IFNβ was considerably lower in the aged animals [50]. In two studies with SARS patients, no expression of IFN or ISGs was detected in peripheral blood mononuclear cells derived from the patients, indicating that these genes were suppressed [51,52]. In one MERS case with a fatal outcome, RIG-I, MDA5 and IRF3/7 were downregulated compared to the levels of these IFN components in a recovering patient, and additionally, no IFNα was produced [53]. 

Several knockout mice models also illustrate the importance of IFN signalling in the control of coronavirus infection. STAT1^-/-^ 129 mice, which are completely deficient for type I, II and III IFN signalling, were more susceptible to a mouse-adapted SARS-CoV (rMA15) and the SARS-CoV Urbani strain and showed more severe pathology [54,55]. In C57BL/6 mice lacking IFNAR1 or IFNLR1, the replication of SARS-CoV in the lungs was enhanced, and in double-knockout mice, the viral titres in the lungs were even further increased [56]. Additionally, mice deficient for both IFNAR1 and IFNLR1 or STAT1 showed delayed clearance of rMA15 from the lungs [57]. Remarkably, IFNAR1^-/-^, IFNLR1^-/-^ [55] and IFNAR1^-/-^/IFNLR1^-/-^ (double-knockout) mice [57] showed no difference in clinical outcome compared to wild-type (wt) SARS-CoV-infected mice, while STAT1^-/-^ mice did present with signs of disease and decreased survival [55,57]. This would suggest that the increased replication of SARS-CoV is not directly related to pathology and disease outcome. In addition, these results highlight that an interferon-independent signalling pathway through STAT1 is possibly important in restricting SARS-CoV pathology. Moreover, C57BL/6J mice deficient in MyD88, TRIF or TLR3 showed more weight loss and more severe lung pathology due to SARS-CoV infection [58,59]. 

Since mice are not naturally susceptible to MERS-CoV, one way to allow MERS-CoV infection is to transduce them with an adenovirus expressing the human DPP4 receptor [60]. Transduced IFNAR1^-/-^ mice showed increased body weight loss, delayed viral clearance and increased inflammation, but these animals did not die from MERS-CoV infection. In the same study, transduced MyD88^-/-^ and MAVS^-/-^ mice were infected with MERS-CoV, which resulted in considerable weight loss in the MyD88^-/-^ group [60]. In summary, these findings from studies in mouse models suggest that the IFN response is important for the clearance and control of coronaviruses during infection. However, the restriction of viral replication does not always seem to be directly associated with the clinical outcome of the infection.

Paradoxically, there is also clear evidence for an immunopathogenic role of IFN in coronavirus infections. The innate immune response and, in particular, the elevated levels of pro-inflammatory cytokines have been considered as possible contributors to the pathology of SARS for a long time [61,62]. High levels of pro-inflammatory cytokines such as IL-1, IL-6 and IFNγ and of chemokines such as CCL2, CXCL10 and IL-8 were detected in the sera of SARS patients [63,64,65]. In another cohort of SARS patients, high expression levels of ISGs such as CD58, IFNAR1 and IFNGR1 (interferon gamma receptor 1) and of the IFN-stimulated chemokines CXCL10 and CCL2 were correlated with disease severity. Especially, CXCL10 was upregulated during the crisis phase in SARS patients who died. Simultaneously, other ISGs were significantly downregulated in severe-SARS patients [66]. This is further supported by studies in MERS patients. Upregulation of CXCL10 has also been demonstrated in the serum of MERS subjects with pneumonia [67]. Correspondingly, Kim et al. showed that in MERS patients, CXCL10 was correlated with severe disease and, in addition to that, its expression was delayed [68]. Moreover, IFNα was found to be associated with severe disease. However, this might be misleading, as patients received PEGylated IFN-α2a, which might have affected the detected association [68]. 

The controversial role of IFN in disease pathogenesis has also been demonstrated in mouse models. In one study, BALB/c mice were infected with SARS-CoV rMA15, and survival and disease severity were measured [41]. BALB/c mice are highly susceptible to this mouse-adapted SARS-CoV, and infection is lethal in young mice [69]. MA15 infection resulted in clear signs of disease in the mice and considerable weight loss, while 85% of the mice died 8 d after infection. Importantly, this study showed that the expression of IFNβ and other pro-inflammatory cytokines was delayed in these severely ill mice. Moreover, all the BALB/c IFNAR^-/-^ mice infected with MA15 survived and only had mild to moderate weight loss and disease scores [41]. In line with that, DeDiego et al. showed that the inhibition of NF-κB resulted in reduced inflammation and lung pathology in BALB/c mice that were infected with MA15 [70]. Together, these observations indicate that a dysregulated and/or delayed IFN response is at least partly responsible for the pathology of SARS and MERS.

### 3.3. Innate Immune Evasion and Coronavirus Pathogenesis

Ambiguously, the IFN response seems to be contributing simultaneously to protection against viruses and to the pathology induced by the same virus infections. Evasion of the IFN response by coronaviruses may thus not necessarily lead to a more severe pathology. Many of the nsps, structural proteins and accessory proteins encoded by coronaviruses have been suggested to antagonize the IFN response. Interestingly, it was shown that in cell culture, MERS-CoV is more sensitive to IFNα and IFNβ treatment than SARS-CoV, which suggests that SARS-CoV is more effective in evading the innate immune response [71,72]. 

In regard to the structural proteins, the role of the E protein in pathogenesis has been studied in particular. DeDiego et al. demonstrated that BALB/c mice infected with MA15 lacking the E protein showed reduced lung pathology, did not suffer considerable weight loss, nor any mortality was observed compared to wildtype MA15-infected mice. The researchers were able to link this reduced virulence of the SARS-CoV-ΔE to a reduced induction of NF-κB [70]. This then suggests that the pathogenesis of SARS-CoV is closely linked to the inflammatory response. Furthermore, an additional study showed that the pathogenesis caused by the SARS-CoV E protein is connected to its ion channel (IC) activity. The authors claim that the IC activity could activate the inflammasome, which would then result in damaging immune responses [73]. 

The accessory genes 3b and 6 of SARS-CoV have been implicated in antagonizing the IFN pathway [74], while other accessory genes still have unknown functions. However, no convincing evidence on the effect of the accessory proteins on SARS-CoV virulence has been generated yet. Zhao et al. only showed a minimal effect of the deletion of accessory protein 6 (p6) on the virulence of SARS-CoV [75]. Other studies found no effect on virulence or virus replication when deleting one or more of the accessory genes [76,77,78]. In contrast, MERS-CoV accessory genes have been shown to influence coronavirus replication and disease outcome in mice. In vitro, MERS-CoV, which lacks genes 3, 4a, 4b and 5, showed an enhanced type I and III IFN response [79]. This virus was then evaluated in a mouse model that was made susceptible to MERS-CoV by modulating DDP4 with CRISPR (288–330^+/+^ mice) [80]. The MERS-CoV mutant replicated less efficiently in the lungs of these mice. Removing the accessory genes 3–5 from a mouse-adapted MERS-CoV, which can cause clinical symptoms, prevented weight loss of the mice and reduced the production of pro-inflammatory cytokines such as IL-6, TNFα and CCL2, although the effect on IFN production was not described by the authors [79]. Moreover, using a chimeric mouse hepatitis virus expressing MERS-CoV accessory protein 4b, it was established that protein 4b is necessary for optimal replication of the virus in primary mouse bone marrow-derived macrophages (BMM). Moreover, when the catalytic site of 4b was mutated, the chimeric virus was able to replicate to titres equivalent to those of the wild-type virus in RNase L^−/−^ BMM and RNase L^−/−^ mice. This indicates that MERS-CoV 4b counteracts the RNaseL pathway, an interferon-induced antiviral pathway, and thereby enhances viral replication [81]. In summary, none of the SARS-CoV accessory proteins have been linked to coronavirus pathogenesis yet. On the contrary, MERS-CoV accessory proteins have been implicated as major virulence factors, possibly by counteracting IFN-induced responses. This difference between the viruses might not be completely surprising, since there is no sequence homology between the accessory genes. However, it is surprising that MERS-CoV seems to counteract the innate immune response in more ways than SARS-CoV, since the observed difference in sensitivity to IFN treatment suggests that SARS-CoV is more effective in evading the innate immune response. This, however, might be explained by the specific IFN pathways that are targeted. For example, SARS-CoV was shown to block STAT1 translocation using p6 [82], while MERS-CoV did not inhibit STAT1 translocation after cells were treated with 1000 ng PEG-IFN ml^−1^ [71]. Thus, the type of pathways blocked by the different coronaviruses might explain differences in sensitivity to IFNs. However, until we have a complete picture of the evasive activities of each of these viruses, we cannot draw firm conclusions on this.

Many of the coronavirus nsps have been implicated in the evasion of innate immunity, but the effects of only a few have been studied in vivo. Nsp3 of coronaviruses encodes a macrodomain which functions as ADP-ribose-1′-phosphatase. Fehr et al. created SARS-CoV mutants that lack this macrodomain activity and infected BALB/c mice with these mutant viruses. Mice survival was significantly increased, and the mice did not develop lung pathology after infection with viruses lacking the macrodomain activity. Moreover, they found increased expression of type I IFN, ISG15, CXCL10 and the inflammatory cytokines IL-6 and TNF [83]. This is interesting, since most studies observed decreases in pro-inflammatory cytokines when survival and disease pathology improved. Remarkably, infection of IFNAR^-/-^ mice did not restore the pathogenesis of the mutant viruses, which might suggest that the type III IFN system or other possibly unknown systems (also) plays an important role in SARS pathogenesis [83]. Besides its macrodomain, nsp3 also contains a papain-like protease (PL^pro^) required for the processing of the coronavirus nsps. PL^pro^ is thought to antagonize the IFN response through its deubiquitinating enzyme (DUB) activity, which allows the protease to remove ubiquitin from a substrate. Ubiquitination is a posttranslational modification that, among many other functions, is very important in the regulation of innate immune pathways [84]. Our group recently tested recombinant MERS-CoV mutants lacking DUB activity but with intact proteolytic activity, in a lethal mouse model. Transgenic mice expressing human DPP4 which were infected with DUB-negative MERS-CoV showed significantly increased survival compared to wt virus-infected mice. Moreover, the levels of IFNβ, IFNλ and several ISGs and pro-inflammatory cytokines were found to be upregulated at an earlier time point in the mice infected with DUB-negative MERS-CoV than in the wt virus-infected mice, while the late induction of the innate immune response in wt virus-infected mice reached an out-of-control high level. This suggests that the DUB-negative MERS-CoV is less effective in evading innate immunity and that the resulting early and balanced immune response is important for controlling the infection, while the delay in innate immune induction caused by the DUB function during wt virus infection dysregulates the response, causing more severe, immunopathological symptoms (unpublished data)[85]. In summary, this indicates that the DUB activity of nsp3 is a major contributor to the pathogenesis of coronavirus infections. Altogether, these studies showed that nsp3 has several mechanisms to evade the immune response, which enhance the pathology of coronaviruses. 

Nsp16 has 2′-O-methyltransferase activity and is thereby involved in capping the viral RNA to prevent detection by innate immune sensors. A SARS-CoV nsp16 mutant without its 2′-O-methyltransferase activity was used to infect BALB/c mice. In comparison to wt infected animals, mice generally suffered less weight loss, had less pronounced lung disease, and their survival rate was increased. Interestingly, the virus was not attenuated anymore in MDA5 knockout mice and in interferon-induced protein with tetratricopeptide repeats (IFIT) family knockout mice, while in the IFNAR1^-/-^ mice, the pathogenesis of the mutant virus was not completely restored [86]. This again suggests a possible role of type III IFNs or of IFIT, which can recognize unmethylated RNA and which might act in an IFN-independent matter in SARS pathogenesis. Moreover, a MERS-CoV nsp16 mutant lacking 2′-O-methyltransferase activity showed attenuated replication in primary human airway epithelial cells and in 288–330^+/+^ mice. This could be attributed to increased recognition by innate immunity, which is supported by an increased sensitivity of the nsp16 mutant to type I IFNs in vitro [87], though a direct effect on virus replication cannot be excluded and might also contribute to the attenuated phenotype in vivo. The nsp16 mutation in a highly lethal mouse-adapted MERS-CoV rescued the mice, and only minimal disease was observed [87]. Together these studies implicate that nsp16 is an important factor in SARS and MERS pathogenesis. The pathogenesis associated with nsp16 seems to be dependent on IFIT and not necessarily on type I IFNs, although this was not tested in a mouse model for MERS-CoV.

In addition to proteins, the coronavirus genome also encodes non-coding RNAs (ncRNAs). The inhibition of a small ncRNA derived from the N gene was associated with decreased inflammation and reduced gross pathology of the lung, though viral titers and survival were not influenced. After pre-treating mice with an inhibitor of the ncRNA-N, the lungs of BALB/c mice infected with MA15 showed increased expression of ISGs (ISG15, MX1) and decreased expression of pro-inflammatory cytokines like IL-6 and CCL2 [88]. This study thereby supports a view in which a dysregulated IFN response and increased inflammation (partly) due to immune evasive activities of the virus are responsible for the lung pathology observed during coronavirus infection.

## 4. Alphaviruses

### 4.1. Genome Organization

Chikungunya virus is an alphavirus, member of the Togaviridae family, which causes fever, skin rash and severe joint pain. The symptoms resolve within weeks, but the arthralgia is recurrent in 30–40% of the cases and can persist for months or years [89]. The virus is transmitted to humans by *Aedes* mosquitoes and currently mainly affects individuals in the Americas. 

Alphaviruses have a genome of around 12kb containing a 5’-terminal cap and a 3’-terminal poly(A) tail (Figure 3). The genome consists of two ORFs; the 5’-terminal ORF encodes the nonstructural proteins, and the 3’-terminal ORF encodes the structural proteins. Consequently, two polyproteins are produced which are cleaved by viral and host proteases into four nonstructural proteins (nsps) and five structural proteins (C, E3, E2, 6K, E1). The nsps are expressed from the genomic RNA and are involved in viral replication. The structural proteins are expressed from a subgenomic RNA and are the essential components of the new viral particles [90,91].

### 4.2. The Role of Interferons in Alphavirus Pathogenesis

The IFN response is also implicated in controlling CHIKV infection. Patients infected with CHIKV had high serum levels of IFNα, and the expression level was positively correlated with the viral load [92]. Besides Chikungunya patients, also mice and non-human primates had high levels of type I IFN in their blood after CHIKV infections [93,94]. However, striking differences in immune profiles were found in patients during each phase of CHIKV infection. During the acute phase, IFNα and several pro-inflammatory cytokines and chemokines peaked, while during the early convalescent phase, T cell cytokines were mainly detected. In the chronic phase, IL-17, a pro-inflammatory cytokine produced by T cells, was significantly upregulated [95]. In patients with persistent joint pain, and thus more severe disease, high levels of IL-6 were detected [95,96]. Moreover, Teng et al. performed a meta-analysis in which they demonstrated that a common expression profile of IFN and pro-inflammatory cytokines was found during the acute phase of CHIKV infection in all patient cohorts [97]. Overall, these studies indicate that in response to CHIKV infection, the IFN system is rapidly activated. 

Treatment with IFNα before infecting mice with CHIKV decreased viremia and disease signs. However, treating the mice on day 3 during the infection did not result in a therapeutic effect [93]. This suggests that during the acute phase, the antiviral response is activated by CHIKV and is involved in controlling CHIKV infection, while in later phases, this response might not be effective. This could be due to the IFN response normally shutting off in this later phase, when CHIKV has already disseminated and caused pathology, thus needing a different kind of immune response to control the infection. Additionally, the IFN response may normally be effectively evaded by CHIKV in the early stages of infection, and pre-treatment may overrule that evasion [93]. Moreover, a study in aged macaques showed that CHIKV infection persists in these elderly animals. This was linked to a reduced ISG response. Rhesus macaque fibroblasts that were stimulated with the sera of aged macaques, collected 3 d post infection, produced lower levels of ISGs than after stimulation with the sera of younger adult animals [98]. On the contrary, persistence of the inflammatory response could also contribute to the pathogenesis of Chikungunya, as demonstrated by the high levels of pro-inflammatory cytokines and chemokines in patients with persistent joint pain [95]. In addition, cellular immune components such as macrophages and CD4^+^ T cells have also been implicated in the immunopathology of the disease [99]. 

The role of innate immunity in CHIKV infection has been extensively studied in experimental animal models. Primary fibroblasts isolated from wt and TLR3^-/-^ mice showed that the replication of CHIKV was increased in the TLR3^-/-^ cells. Interestingly, high levels of type I IFN were expressed in TLR3^-/-^ fibroblasts. This suggests that other PRRs activate the IFN response, although this was not sufficient to control replication, since replication was enhanced in TLR3-deficient cells. In TLR3^-/-^ C57BL/6 mice, the viral load was increased, inflammation was more severe, and the virus could disseminate more extensively [100]. This is supported by another study that found TRIF^-/-^ mice had an increased viral load and inflammation (foot swelling) compared to wt mice infected with CHIKV [101]. However, conflicting results were obtained with TLR3^-/-^ mice in a study by Schilte et al. [92]. These authors did not observe any changes in viral load in tissue or blood, though Her et al. [100] suggested that this might be related to methodology differences such as the age of the mice and the injection route used. Furthermore, CHIKV infection is not lethal in wt immunocompetent C57BL/6 or 129 mice. However, IFNAR^-/-^ and STAT^-/-^ mice do succumb to the infection, and the pathology is more severe [92,102,103]. This suggests an important role for IFN in CHIKV pathogenesis. However, it was striking that in one of these studies, the wt mice did not produce detectable protein levels of type I IFN during infection [92]. Moreover, RIG-I^-/-^, MDA5^-/-^, MAVS^-/-^ or MyD88^-/-^ mice infected with CHIKV survived, and only small increases in viral load in their sera or tissues were observed. MyD88^-/-^ mice did show a slightly more severe phenotype with dissemination of CHIKV 72 h post infection. The results together probably indicate that several PRRs are involved in sensing CHIKV and that missing one of the receptors is not sufficient to induce a lethal phenotype, as seen in the IFNAR^-/-^ mice [92]. In line with this, Rudd et al. showed increased viremia in MAVS^-/-^ and MyD88^-/-^ mice [101]. Moreover, in agreement with earlier studies, they showed that IFNAR^-/-^ mice died and had a severe pathology (i.e., oedema and haemorrhagic shock). In addition, Rudd et al. evaluated the influence of IRF3, IRF7 and IRF3/7 on chikungunya pathology. All IRF3^-/-^ and IRF7^-/-^ mice survived CHIKV infection, while the double-knockout mice all died within 6 days. Viral titres in both blood and tissues were significantly higher for the IRF3/7^-/-^ mice compared to the wt mice. In all three groups of knockout mice, disease symptoms occurred earlier and were more severe compared to wt mice. Furthermore, in IRF3/7^-/-^ mouse serum, high levels of pro-inflammatory cytokines and no IFNα/β were detected. IRF7 was found to be most important for the induction of IFN after CHIKV infection [101]. Lastly, several ISGs have also been shown to influence the viral titres in blood or tissues [104,105,106] or survival [107]. From these different knockout models, it can be concluded that the IFN response greatly influences disease control and severity and that CHIKV is sensed dominantly through TLR3, though adaptive immunity has also been shown to contribute to the control of CHIKV infection [108]. 

### 4.3. Innate Immune Evasion and Alphavirus Pathogenesis

CHIKV is thought to evade innate immune responses by causing a transcriptional and translational shut-off in the host cells [109]. Chikungunya infection triggers IFNβ and ISG mRNA production that is dependent on IRF3 in primary human fibroblasts. However, no protein is expressed during infection, illustrating that CHIKV blocks host cell translation thereby preventing the ISG effector response. Moreover, a study showed that 24 h post infection, transcription is also effectively inhibited [110]. NsP2 of CHIKV is thought to be involved in mediating the transcriptional host shut-off, although this has only been established in cell lines. For Sindbis virus, a relative of CHIKV, a nsP2 mutant was shown to increase IFN production, and survival of the infected mice was considerably increased [111]. The exact mechanisms of the host translational shut-off and the viral proteins involved are not clear yet, which hampers the investigation of the role of this evasion method in Chikungunya pathogenesis. In addition, nsP2 of alphaviruses has been suggested to directly inhibit STAT signalling, even though this has not been investigated in experimental animal models yet [109]. 

Recently, Chan et al. [112] studied the role of nonstructural protein mutations in CHIKV pathogenesis. A mutation in the nsP1/2 cleavage site (the mutant was called RH) that affects polyprotein processing resulted in reduced viremia; the virus was cleared earlier from the blood and caused reduced inflammation. This also correlated with locally increased IFNα/β and pro-inflammatory cytokines and chemokines levels 15 h post infection. A second mutant with a substitution in the protease domain of nsP2 (mutant EV) actually showed increased viremia and joint inflammation compared to wt CHIKV. A double mutant containing both mutations (mutant RHEV) behaved like the RH mutant. Interestingly, wt and EV CHIKV showed a delayed induction of inflammatory cytokines and chemokines compared to RH and RHEV virus, and 6 d post infection, these viruses produced higher levels of pro-inflammatory cytokines like IL-6 and IL-1 than the RH and RHEV mutants [112]. This supports the notion that the antiviral IFN responses affect disease severity and again shows that a dysregulated innate immune response, inflicted by viral immune evasion activities, might contribute to pathology. Overall, CHIKV nsP2 has often been suggested as an important viral factor involved in innate immune evasion and might thus be a major cause of pathology, though this has not yet been confirmed in vivo. Therefore, much remains to be discovered about the immune evasion by CHIKV proteins in the context of viral infection.

## 5. Flaviviruses 

### 5.1. Genome Organization

Flaviviruses comprise human pathogens that are the main cause of arthropod-borne diseases [113]. This review will place emphasis on DENV and the recently emerged ZIKV, since these currently have the largest societal impact. Infections by both these viruses are mainly asymptomatic. Dengue is endemic in many countries in the Americas, Southeast Asia and Africa. There are five distinct serotypes of DENV [114]. The clinical symptoms of Dengue include fever, skin rash and headache and, in severe cases, the infection will lead to plasma leakage with or without haemorrhage. The disease is mostly self-limiting, but a small proportion of patients will develop severe disease [115,116]. ZIKV is most prevalent in the Americas and, in mild cases, it causes fever, rash, muscle pain and headache. ZIKV can cause neurological complications such as the Guillain–Barré syndrome and, especially during the early stages of pregnancy, infection leads to congenital abnormalities such as microcephaly [116,117]. Both ZIKV and DENV are transmitted by the *Aedes* mosquito. 

Flaviviruses have a genome of around 11 kb containing a 5’-terminal cap, but they lack a poly(A) tail (Figure 4). The genome encodes one large polyprotein that is processed by viral and host proteases into the three structural proteins (E, C and prM) and the seven nonstructural proteins (NS1, NS2A, NS2B, NS3, NS4A, NS4B and NS5). The nonstructural proteins are involved in viral replication and modulation of host responses, while the structural proteins comprise the components that are assembled into new viral particles [113]. 

### 5.2. The Role of Interferons in Flavivirus Pathogenesis

The IFN response is rapidly upregulated in Zika patients. Zika patients in the acute phase of infection had increased mRNA levels of IFNα, IFNβ and IFNγ in their peripheral blood compared to healthy controls, and ZIKV infection was also associated with high mRNA levels of TLR3. In the same study, it was found that the viral load was positively correlated with type I IFN expression [118]. However, some pro-inflammatory cytokines and IFN-stimulated chemokines have been associated with severity of both Zika and Dengue. One study showed that Zika patients with moderate symptoms and viremia had higher levels of IL-1RA, CCL2, CXCL10 and IL-8 than patients with mild symptoms or no viremia [119]. CCL2 and CXCL10 have also been associated with ZIKV-infected women carrying foetuses with congenital abnormalities [120]. In Dengue patients, the severity of the disease has been associated with high plasma levels of IL-1RA and CXCL10 [121]. Moreover, severe cases of Dengue with increased levels of pro-inflammatory cytokines such as IL-1 and TNFα, anti-inflammatory cytokine IL-10 and chemokines including IL-8 and CXCL10 have been reported [122,123,124,125]. This cytokine storm in dengue patients can lead to the plasma leakage and haemorrhage seen in severe Dengue cases. This implicates that immunopathology is a substantial part of flavivirus pathogenesis.

The importance of interferons in restricting DENV and ZIKV infections is quite clear from mouse models, since only mice lacking innate immune components are susceptible to replication of these flaviviruses. After subcutaneous infection of ZIKV in adult immunocompetent mice, the mice did not show any sign of disease, and viral titres were not detected in the blood of these mice. However, A129 mice that lack the type I IFN receptor did show signs of illness and had measurable blood titres. Lethality in A129 mice was shown to be age-dependent. In 3-week-old A129 mice, severe disease was observed, and all animals succumbed to the ZIKV infection, while 5-week-old mice had reduced mortality, and 11-week-old A129 mice all survived [126]. In contrast, Dowall et al. showed that 5–6-week-old A129 mice all died from ZIKV infection [127], although this difference might be explained by the virus strain and dosage. Three-week-old AG129 mice, which are deficient in type I and II IFN signalling, also died after ZIKV infection. No significant differences in mortality rate, weight loss and organ viral titres were found between the A129 and the AG129 mice [126]. This indicates that especially the type I IFN response is important in controlling ZIKV infection. However, Aliota et al. showed that both young and adult AG129 mice succumbed to ZIKV infection [128], which might highlight a role of IFNγ in controlling ZIKV infection in adults and could possibly explain the age-dependent mortality in A129 mice observed by Rossi et al. [126,129]. AG129 mice also succumbed to infection with DENV (serotypes 1 and 2), though A129 mice all survived this infection [130,131]. A129 mice did have disseminated infection of DENV2 [130]. Therefore, in contrast with ZIKV infection, both type I and II IFN responses appear to be important in restricting DENV infections in mice. IFNAR^-/-^ C57BL/6 mice also died from infection with two mouse-adapted DENV2 strains (D2S10 [132] and D220) and had measurable viral titres in blood and tissues [133]. Blocking of the IFN I receptor with an antibody mimicked the phenotype seen in knockout mice. In wildtype C57BL/6 ZIKV-infected mice, treatment with a blocking antibody resulted in 40–100% mortality depending on the route of ZIKV injection and induced severe neuropathology. Comparable to what observed in the A129 mice, the blocking antibody also resulted in detectable viral titres in blood and various tissues [134]. 

In accordance with these observations, flavivirus infection and its control have also been associated with signalling components of the IFN pathways. For example, wildtype C57BL/6 mice were infected intravaginally with ZIKV, and it was shown that ZIKV was able to replicate in the lower female reproductive tract. The measured viral load was positively correlated with the induction of IRF7. Moreover, treatment of these mice with acitretin (an enhancer of RIG-I signalling) one day prior to infection, induced type I and III IFNs and pro-inflammatory cytokines, while inhibiting ZIKV replication in the lower female reproductive tract [135]. C57BL/6 mice lacking IFNAR1, MAVS, IRF3 or IRF3/5/7 were inoculated with ZIKV. The IFNAR^-/-^ and the IRF triple-knockout mice significantly lost weight and succumbed to disease, although the percentage of surviving mice was strain- and age-dependent. MAVS^-/-^ and IRF3^-/-^ mice showed no morbidity or mortality [136]. C57BL/6 mice deficient in IRF3 and IRF7 developed viremia, while wt mice did not [137]. Mice deficient in MAVS, TRIF or both (DK) that were infected with ZIKV exhibited no signs of disease or mortality, though the virus was detected in the blood and tissues. The viral titre in the blood and tissues was the highest for DK mice. In addition, the higher viral loads were associated with decreased type I IFN production and IRF7 expression in the spleen [138]. In DENV-infected C57BL/6 mice lacking the MAVS adapter, no mortality was observed either, though viremia was detected and delayed, and decreased type I IFN expression in blood and spleen was observed, as was seen with the ZIKV-infected mice [139]. DENV2 infection induced lethality in STAT^-/-^ BALB/c mice [140]. Lastly, a combination of TLR3 and TLR7/8 agonists was shown to protect rhesus macaques against DENV infection by decreasing viremia, though the study groups were very small [141]. 

The results on the role of type I IFNs in ZIKV pathogenesis during pregnancy are conflicting. There is evidence that links type I IFNs to protection of the placenta and foetus against ZIKV. For example, the mid-gestation placenta is largely resistant to ZIKV infection, and this is associated with higher levels of IFNα, IFNβ and IFNλ expression [142]. Moreover, to study the effect of ZIKV infection during pregnancy, a trans-placental transmission mouse model of ZIKV has been developed by knocking out IFNAR1. Using this model, both protective and harmful effects of IFN were found. Crossing IFNAR^-/-^ female mice with male wt mice resulted in IFNAR^-/+^ placenta and foetuses. ZIKV infection of these IFNAR^-/+^ mice at E10.5 (mid-pregnancy) did not result in foetal demise and caused reduced damage to the placenta compared to ZIKV infection of foetuses completely deficient in type I IFN signalling. However, the foetuses still suffered from intrauterine growth retardation [143]. This study mainly suggests that the presence of IFN signalling in the foetuses and placenta had a protective effect. In contrast, when pregnant IFNAR1^-/-^ female mice were mated with wt male mice and infected with ZIKV at E6.5 and E7.5, foetal demise occurred in the majority of the cases. The rest of the IFNAR1^+/-^ foetuses exhibited intrauterine growth retardation. Moreover, treating pregnant wt mice with an IFNAR1 blocking antibody also resulted in growth retardation, although to a lesser extent than in the IFNAR1^+/-^ foetuses [144]. Yockey et al. [145] additionally showed that IFNAR1^-/-^ foetuses did not undergo foetal demise when the pregnant dams were infected with ZIKV at E5.5. When dams got infected at E8.5 with ZIKV, IFNAR1^-/-^ foetuses showed reduced intrauterine growth retardation compared to the IFNAR1^+/-^ foetuses. Viral titres were comparable to those in the IFNAR^+/-^ placentas or higher than those in the IFNAR^-/-^ placentas, indicating that the type I IFN response, and not the virus load, is responsible for a more severe disease phenotype. The authors also examined the effect of a treatment with recombinant IFNβ and IFNλ3 on human chorionic villous isolated from mid-gestation placentas. IFNβ treatment resulted in abnormal villous structures, while IFNλ treatment did not influence the villi [145]. Interestingly, there was a huge difference in genes induced by the two types of IFN, although they share the same signalling pathways. The conflicting results about the role of IFN might be partially explained by the differences in the embryonic days analysed. It is however clear that the role of type I IFNs in protection of the foetus against ZIKV requires further investigation. 

A protective role of type III IFNs in restricting ZIKV during pregnancy has been clearly identified. Primary human placental trophoblasts have been shown to resist ZIKV infection and produce high basal levels of type III IFN [146]. Moreover, two 3D models consisting of primary placental fibroblasts or endothelial cells co-cultured with JEG-3 cells that better reflect primary human trophoblasts were developed. Using these models, it was shown that 3D-cultured JEG-3 cells became less susceptible to ZIKV, while expressing high basal levels of IFNλ1/2 and ISGs. In addition, human placental villi isolated from second-trimester placentas, which are mostly resistant to ZIKV infection, were also capable of producing IFNλ1 [147]. Furthermore, evidence from mice models also supports a protective role of type III IFNs. Mice deficient in IFNLR1 were mated, and the pregnant dams were infected with ZIKV at E12 (embryonic day 12) after pre-treatment with an antibody blocking IFNAR1 to allow trans-placental transmission. ZIKV titres in the placenta and foetal heads were higher in IFNLR1-deficient mice and foetuses than in wt mice. Interestingly, IFNLR1^+/-^ placentas and foetuses had ZIKV titres comparable to those of wt mice, indicating that type III IFN in the placenta and foetus is sufficient to control ZIKV. This is further supported by the fact that pegylated mouse IFN-λ2 treatment of wt mice (pre-treated with an IFNAR1 blocking antibody) infected with ZIKV at E12 presented with reduced ZIKV titres in the placenta and in the foetal heads. Furthermore, isolated human chorionic villi, foetal membranes and decidua basalis all responded to IFNλ1 and IFNλ3 treatment by inducing ISGs such as OAS1 and had reduced ZIKV titres [148]. A second study also showed the beneficial effects of IFNλ2 treatment. Treating ZIKV-infected IFNAR^-/-^ pregnant mice with type III IFNs at E10.5 prevented foetal demise and improved foetal growth. In addition, ZIKV titres were reduced in both maternal and foetal organs [143]. Both studies found no beneficial effect of IFNλ treatment early during pregnancy (administered at E5.5 and E6).

### 5.3. Innate Immune Evasion and Flavivirus Pathogenesis

In particular, the nsps of the flaviviruses are considered antagonists of host innate immunity. NS1 of DENV has been shown to increase pathogenesis in mice. Sublethal doses of DENV2 in combination with recombinant NS1 resulted in lethality in IFNAR^-/-^ mice, and NS1 was shown to induce vascular leakage. Moreover, when these mice were vaccinated over 6 weeks with DENV2 NS1 and later challenged with a lethal dose of DENV2, none of the mice died [149]. However, since these results were obtained in IFNAR^-/-^ mice, the link with innate immune evasion cannot be established. The immunocompromised mouse models currently used will always limit studies addressing dengue pathogenesis in the context of innate immunity, and therefore the development of a more adequate animal model is desirable. 

DENV NS5 is thought to inhibit IFN signalling by degrading STAT2 [150]. Interestingly, NS5 has been shown to confer host specificity. For example, NS5 cannot inhibit mouse STAT2 in several cell lines, and this is suggested to be one of the main reasons why DENV is not able to replicate in immunocompetent mice. This hypothesis is supported by the fact that STAT2^-/-^ mice led to delayed clearance of DENV [151]. A mouse model with a humanized STAT2 might thus be interesting to study the effect of NS5 and possibly other immune evasion mechanisms on Dengue pathogenesis [151]. 

The flavivirus NS2B3 complex serves as a protease and can cleave human STING in vitro and is thereby thought to inhibit the antiviral immune response and to support viral replication. However, mice deficient in STING (*Tmem173^Gt^*) did not respond any differently to ZIKV infection than wildtype mice [152], which also makes it less likely that NS2B3 will affect Zika pathogenesis through its action on STING. However, NS2B3 might still target other signalling molecules to interfere with the innate immune response. 

Another study showed that the inflammasome can decrease the IFN response and support viral replication [153]. This was established using Nlrp3^-/-^ neonatal mice (Nlrp3 plays a role in inflammasome activation) that were injected with ZIKV in the brain. The knockout mice showed higher expression of IFNβ and lower viral titres in the brain and gained more weight than the wt mice. In human peripheral mononuclear blood cells, it was also shown that Nlrp3 inhibition increased IFNβ expression. The same study showed that NS1 can activate the inflammasome in cell lines [153]. Together, these data suggest that ZIKV NS1 might be able to support viral replication through the activation of the inflammasome, but this needs further confirmation.

Xia et al. [154] discovered that ZIKV NS1 can suppress IFNβ in vitro dependently on the ZIKV strain (FSS13025 or PRVABC-59). NS1 of FSS13025 was not able to suppress IFNβ, in contrast to NS1 of PRVABC-59. There was only one amino acid difference in NS1 between the two strains, i.e., an A188V substitution. The substitution was shown to be responsible for suppressing IFNβ expression in bone marrow-derived dendritic cells (BMDCs) isolated from C57BL/6J mice [154]. Moreover, Xia et al. infected A129 mice with the two strains of ZIKV and with two mutant viruses, FSS13025 A188V and PRVABC-59 V188A. The viruses with an alanine at position 188 in NS1 showed considerably higher levels of IFNβ in the sera of infected animals. However, the survival of neonatal CD-1 mice infected with a mutant virus was not significantly different compared to the survival of wt-infected mice [154], suggesting that the effect of NS1 alone is not a significant contributor to the pathogenesis of ZIKV. The same authors demonstrated that NS2A, NS2B, NS4A, NS4B and NS5 were able to suppress IFNβ in vitro, which leaves many potential candidates to investigate the virulence of ZIKV. 

## 6. Conclusions

Although the IFN response is regarded as the first line of defence against viruses, its role in the pathogenesis of viral infections is double-sided. On the one hand, many +ssRNA viruses have developed mechanisms to antagonize the IFN response to protect themselves against the antiviral effects, and without a proper immune response that restricts virus replication, the infection as well as the disease might progress. On the other hand, viral evasion often causes delayed, hyper-induced IFN and inflammatory responses, which have been shown to cause pathology. Animal models help us to assess the function of immune evasion viral proteins in the context of virus infection and their clinical outcomes. Conflicting results on the link between IFN pathways and viral pathogenesis remain, which probably emphasize this ambiguous role of the innate immune response in pathogenesis. Overall, from these studies, we can conclude that a balance and a correct timing of the IFN response seem most important in controlling disease severity. Also the importance of type III IFN in controlling virus infections is becoming more evident, as was discussed for ZIKV [15]. Elucidating the role of type III IFN in other viral infections is thus of great interest, especially for viruses targeting mucosal surfaces, such as MERS-CoV. Moreover, as we gain more knowledge on the interaction between innate immunity and viral infection, including the activities of viral immune evasive proteins, possibilities for the development of novel treatments and prevention strategies might appear on the horizon. 

## Figures and Tables

**Figure 1 viruses-11-00961-f001:**
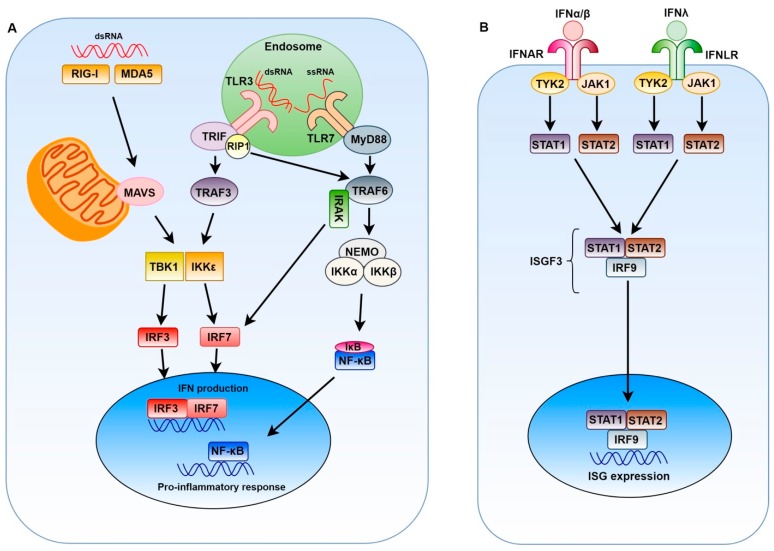
Innate immune response to positive-stranded RNA viruses and interferon (IFN) signalling. (**A**) IFN production is induced when single-stranded RNA (ssRNA) (TLR7, RIG-I) or double-stranded RNA (dsRNA) (TLR3, RIG-I, MDA5) is detected in the cell. Signalling through these pattern recognition receptors (PRRs) will ultimately result in the translocation of NF-κB, IRF3 and IRF7 to the nucleus. These transcription factors then initiate production of IFNs and pro-inflammatory cytokines. (**B**) Type I IFNs bind to a receptor composed of IFNAR1 and IFNAR2, while type III IFNs signal through IFNLR1 and IL10RB. The two types of signalling pathways then converge and result in the expression of interferon-stimulated genes (ISGs).

**Figure 2 viruses-11-00961-f002:**
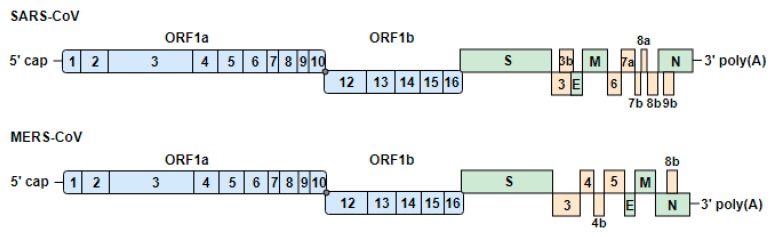
Genome organization of severe acute respiratory syndrome coronavirus (SARS-CoV) and Middle East respiratory syndrome-CoV (MERS-CoV). The genomes encode two large open reading frames (ORF1a and ORF1b), which contain 16 nonstructural proteins (1 to 16). ORF1b is transcribed after a −1 ribosomal frameshift (gray dot). The structural proteins (S, spike; E, envelope; M, membrane; N, nucleocapsid) and accessory proteins are expressed from subgenomic RNAs. Blue, green and yellow indicate the nonstructural, structural and accessory proteins, respectively.

**Figure 3 viruses-11-00961-f003:**
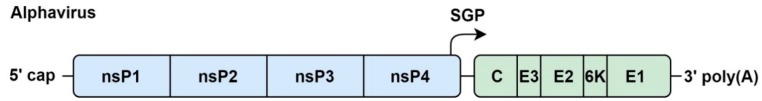
Genome organization of alphaviruses. The genome encodes two ORFs, which contain the four nonstructural proteins (nsP) and the structural proteins (C, capsid; E, envelope). The structural proteins are expressed from a subgenomic promoter (SGP). Blue and green indicate the nonstructural and structural proteins, respectively.

**Figure 4 viruses-11-00961-f004:**
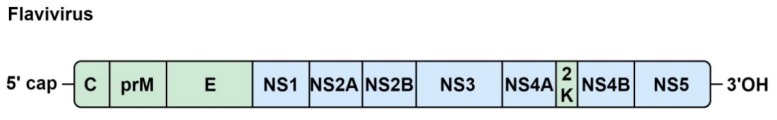
Genome organization of flaviviruses. The genome encodes one ORF, which contains the seven nonstructural proteins (NS) and the structural proteins (C, capsid; prM, premembrane; E, envelope). Blue and green indicate the nonstructural and structural proteins, respectively.

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
