# Peer review of "Viral Innate Immune Evasion and the Pathogenesis of Emerging RNA Virus Infections"

_viruses, 2019, doi:10.3390/v11100961_

Round 1

Reviewer 1 Report

This is a comprehensive and well-written review. This will be very useful for those in the field. I had only minor comments with regard to its readiness for publication. If anything, this review is too comprehensive, sometimes going into unnecessary detail speculating about mechanisms of antagonizing interferon that have not been established. However, in general the manuscript is appropriate for publication with minor revisions.

I am curious why the authors excluded other positive-sense RNA viruses, such as picornaviruses, from this review. Picornaviruses certainly are "emerging RNA viruses" (for example, EV-D68 is implicated in poorly understood outbreaks occurring every two years) and there are defined effects of antiviral immunity on picornaviruses. 

P6—The authors state that the accessory proteins of SARS-CoV have no known functions, including whether they act as interferon antagonists. Yet they then discuss this for a full paragraph, and follow this by noting that accessory proteins of MERS-CoV ARE known virulence factors, though it is unclear if this mechanism is IFN dependent. This entire paragraph section seems unnecessary and somewhat off-topic. The authors could summarize these findings more succinctly and talk about the evidence (if any) that the MERS-CoV proteins are IFN antagonists.

“nsps” is not defined throughout—I realize this means “non-structural protein” but this should be defined at some point in the text

Author Response

Hereby we resubmit a revised version of the manuscript viruses-608268 “Viral Innate Immune Evasion and the Pathogenesis of Emerging RNA Virus Infections”. We would like to thank the reviewers for their time and their useful comments. We have incorporated the suggested revisions to the best of our ability. Below we have addressed the comments of the reviewers point-by-point. Page numbers and lines refer to the revised manuscript with tracked changes.

Reviewer 1: This is a comprehensive and well-written review. This will be very useful for those in the field. I had only minor comments with regard to its readiness for publication. If anything, this review is too comprehensive, sometimes going into unnecessary detail speculating about mechanisms of antagonizing interferon that have not been established. However, in general the manuscript is appropriate for publication with minor revisions.

We would like to thank this reviewer for his/her positive feedback.

I am curious why the authors excluded other positive-sense RNA viruses, such as picornaviruses, from this review. Picornaviruses certainly are "emerging RNA viruses" (for example, EV-D68 is implicated in poorly understood outbreaks occurring every two years) and there are defined effects of antiviral immunity on picornaviruses.

For the scope of this review we decided to limit the evaluated +ssRNA viruses to viruses that are currently of greatest societal impact, a threat to public health, or within the field of our expertise. Moreover, we felt that the current literature evaluating the pathogenesis of recent emerging picornaviruses like EV-D68, which can pose a threat to public health, is not as comprehensive as the literature regarding the reviewed viruses.

P6—The authors state that the accessory proteins of SARS-CoV have no known functions, including whether they act as interferon antagonists. Yet they then discuss this for a full paragraph, and follow this by noting that accessory proteins of MERS-CoV ARE known virulence factors, though it is unclear if this mechanism is IFN dependent. This entire paragraph section seems unnecessary and somewhat off-topic. The authors could summarize these findings more succinctly and talk about the evidence (if any) that the MERS-CoV proteins are IFN antagonists.

We agree that the findings on SARS-CoV could be summarized and that the action of the accessory proteins on the IFN response is not clearly discussed in the original manuscript. The discussion on the SARS-CoV accessory proteins has now been described more concisely and the possible link between MERS-CoV accessory proteins and the IFN response has been outlined. See page 7 lines 258-277.

“nsps” is not defined throughout—I realize this means “non-structural protein” but this should be defined at some point in the text

The term “nsps” has been clarified for the different virus families (if applicable) separately in the text: page 4, line 160, and page 8, line 351.

Reviewer 2 Report

This an interesting and well written review dealing with virus activation, activity and evasion of the interferon systems and the consequence for virus replication and pathogenicity.  Since the review only discusses relatively non-cytolytic (not polio) positive strand viruses, this should be reflected in the title.  Some comments are listed below.

The review discusses active interferon pathway inhibition mechanisms of the viruses delivered by viral proteins.  Mention should be made in an early paragraph that rapidly replicating and cytolytic viruses don't need to inhibit these pathways and that (+) RNA viruses also sequester their viral replication complex in membrane structures which will limit interferon stimulation.  

Line 326. Attenuated replication by a mutant lacking 2'O methytransferase can be due to either a direct effect on virus replication or increased recognition and activation of interferon pathways.  Both should be mentioned as possibilities.

355 typo: envelope

363 grammar: The use of commas should be checked for the next two sentences. 

370: There is no basis yet in the narrative for stating that IFN is responsible for controlling CHIKV just because it is there. This statement would be appropriate after the next paragraph. 

376: Interesting that IFN works early but not late on the infection, sort of like antiviral drugs against Flu.  Alternative possibility is that the virus has already spread, pathogenesis has been induced and other protections must act.

400: Use of 'and' is confusing.  Are all these mutations in one mouse or are these separate. If separate, then use 'or' within the list. 

412: insertion: IRF3/7-/- MOUSE serum

473: Presence of IFN does not mean control since you already comment on the inflammatory pathogenesis that it can induce. 

488: delete 'On the contrary'

493: This is a very interesting observation regarding age-dependent IFN action.  Please provide a suggestion as to why.

504: No proof was provided to substantiate a role for IFN gamma.

Author Response

Reviewer 2: This an interesting and well written review dealing with virus activation, activity and evasion of the interferon systems and the consequence for virus replication and pathogenicity. Since the review only discusses relatively non-cytolytic (not polio) positive strand viruses, this should be reflected in the title.  Some comments are listed below.

We thank this reviewer for his/her positive remarks. We do realise that we have not covered all positive stranded viruses in our review, but to our opinion we have clearly defined the scope of the review in the abstract and in the introduction, and the title does not claim discussion of all positive strand RNA viruses. Therefore we would like the title to remain as it is.

The review discusses active interferon pathway inhibition mechanisms of the viruses delivered by viral proteins. Mention should be made in an early paragraph that rapidly replicating and cytolytic viruses don't need to inhibit these pathways and that (+) RNA viruses also sequester their viral replication complex in membrane structures which will limit interferon stimulation. 

We thank the reviewer for this helpful comment. Information regarding the cytolytic viruses and the replication organelles have been added to the introduction of the review (page 2, lines 45-56).

Line 326. Attenuated replication by a mutant lacking 2'O methytransferase can be due to either a direct effect on virus replication or increased recognition and activation of interferon pathways. Both should be mentioned as possibilities.

Possible explanations for the attenuated phenotype of the nsp16 mutant have been described more thoroughly now. Both the direct effect on virus replication and an increased innate immune response have been mentioned (page 8 lines 326-329).

355 typo: envelope

We thanks the reviewer for spotting this, we found an additional similar typo; thus we have corrected the typo in two places: (legend fig. 2: page 5 line 169, and legend fig. 3: page 9 line 358 of revised manuscript).

363 grammar: The use of commas should be checked for the next two sentences.

The comma placement has been revised (page 9, lines 465-367).

370: There is no basis yet in the narrative for stating that IFN is responsible for controlling CHIKV just because it is there. This statement would be appropriate after the next paragraph.

We agree that this conclusion is drawn too early. The statement has been changed in the first paragraph, and the control of CHIKV by IFN in an early phase is now only mentioned in the second paragraph (page 9, lines 372-378).

376: Interesting that IFN works early but not late on the infection, sort of like antiviral drugs against Flu. Alternative possibility is that the virus has already spread, pathogenesis has been induced and other protections must act.

We thanks the reviewer for this suggestion. The abovementioned possibility has been added to the list of possible explanations (page 9, lines 376-382).

400: Use of 'and' is confusing. Are all these mutations in one mouse or are these separate. If separate, then use 'or' within the list.

In the text “and” has been replaced with “or”(page 9 line 405).

412: insertion: IRF3/7-/- MOUSE serum

“Mouse” has been added to the text and a comma added for clarity (page 10, line 417).

473: Presence of IFN does not mean control since you already comment on the inflammatory pathogenesis that it can induce.

The sentence has been reformulated (page 11, line 532).

488: delete 'On the contrary'

“On the contrary” has been removed from the sentence (page 11, line 493).

493: This is a very interesting observation regarding age-dependent IFN action. Please provide a suggestion as to why.

The type II IFN response is suggested to be effective in controlling ZIKV infection in adult animals, which might also explain the age-dependent effect observed in IFNAR KO mice. Since there is no data as to why this may be the case we did not further comment on it. We did add some extra clarification to the text (page 12, lines 506-508).

504: No proof was provided to substantiate a role for IFN gamma.

Mice deficient in the type I and II IFN receptor (AG129) succumb to DENV infection, while mice lacking only the IFNAR (A129) do survive. In our opinion, this suggests a protective role of IFN gamma in DENV pathogenesis. Since there is no data as to explain the mechanism for this involvement we did not further comment on it. The data was discussed on page 12 lines 508-511.

We hope we have addressed all comments adequately, and the manuscript is now suitable for publication. We thank both reviewers for their helpful input once again.

With best regards,

Tessa Nelemans and Marjolein Kikkert.